# Correlation between Chest Computed Tomography Score and Laboratory Biomarkers in the Risk Stratification of COVID-19 Patients Admitted to the Emergency Department

**DOI:** 10.3390/diagnostics13172829

**Published:** 2023-08-31

**Authors:** Cartesio D’Agostini, Jacopo M. Legramante, Marilena Minieri, Vito N. Di Lecce, Maria Stella Lia, Massimo Maurici, Ilaria Simonelli, Marco Ciotti, Carla Paganelli, Alessandro Terrinoni, Alfredo Giovannelli, Massimo Pieri, Mariacarla Gallù, Vito Dell’Olio, Carla Prezioso, Dolores Limongi, Sergio Bernardini, Antonio Orlacchio

**Affiliations:** 1Department of Experimental Medicine, University of Rome Tor Vergata, 00133 Rome, Italy; d.agostini@med.uniroma2.it (C.D.); alessandro.terrinoni@uniroma2.it (A.T.); massimo.pieri@uniroma2.it (M.P.); bernards@uniroma2.it (S.B.); 2Laboratory of Microbiology, Polyclinic of “Tor Vergata”, 00133 Rome, Italy; 3Department of Systems Medicine, University of Rome Tor Vergata, 00133 Rome, Italy; legraman@uniroma2.it (J.M.L.); mariacarla.gallu@uniroma2.it (M.G.); 4Emergency Department, Tor Vergata University Hospital, 00133 Rome, Italy; vitonicola.dilecce@ptvonline.it (V.N.D.L.); carla.paganelli@ptvonline.it (C.P.); 5Unit of Laboratory Medicine, Tor Vergata University Hospital, 00133 Rome, Italy; lia.mariastella@gmail.com (M.S.L.); alfredo.giovannelli@gmail.com (A.G.); 6Department of Biomedicine and Prevention, University of Rome Tor Vergata, 00133 Rome, Italy; maurici@med.uniroma2.it; 7Nursing Science and Public Health, University of Rome Tor Vergata, 00133 Rome, Italy; ilariasimo@gmail.com; 8Unit of Virology, Tor Vergata University Hospital, 00133 Rome, Italy; marco.ciotti@ptvonline.it; 9Department of Surgical Science, University of Rome Tor Vergata, 00133 Rome, Italy; vito.dellolio@ptvonline.it (V.D.); aorlacchio@uniroma2.it (A.O.); 10Emergency Radiology Unit, Tor Vergata University Hospital, 00133 Rome, Italy; 11Laboratory of Microbiology of Chronic-Neurodegenerative Diseases, IRCCS San Raffaele Roma, 00166 Rome, Italy; carla.prezioso@sanraffaele.it; 12Department of Human Sciences and Quality of Life Promotion, San Raffaele University, 00166 Rome, Italy; dolores.limongi@sanraffaele.it

**Keywords:** emergency department, CT score, laboratory biomarkers, MR-proadrenomedullin

## Abstract

Background: It has been reported that mid-regional proadrenomedullin (MR-proADM) could be considered a useful tool to stratify the mortality risk in COVID-19 patients upon admission to the emergency department (ED). During the COVID-19 outbreak, computed tomography (CT) scans were widely used for their excellent sensitivity in diagnosing pneumonia associated with SARS-CoV-2 infection. However, the possible role of CT score in the risk stratification of COVID-19 patients upon admission to the ED is still unclear. Aim: The main objective of this study was to assess if the association of the CT findings alone or together with MR-proADM results could ameliorate the prediction of in-hospital mortality of COVID-19 patients at the triage. Moreover, the hypothesis that CT score and MR-proADM levels together could play a key role in predicting the correct clinical setting for these patients was also evaluated. Methods: Epidemiological, demographic, clinical, laboratory, and outcome data were assessed and analyzed from 265 consecutive patients admitted to the triage of the ED with a SARS-CoV-2 infection. Results and conclusions: The accuracy results by AUROC analysis and statistical analysis demonstrated that CT score is particularly effective, when utilized together with the MR-proADM level, in the risk stratification of COVID-19 patients admitted to the ED, thus helping the decision-making process of emergency physicians and optimizing the hospital resources.

## 1. Introduction

Among the family of vasoactive peptide hormones associated with the gene of calcitonin, there is a regulatory amino acid peptide, known as adrenomedullin (ADM) [1]. ADM can determine a vasodilating effect through the synthesis of nitric oxide and by increasing the concentration of calcium and cAMP. This process involves an indirect reduction in vascular resistance [2].

ADM can also reduce vascular permeability and the production of pro-inflammatory cytokines and increase natriuresis and diuresis [3,4]. Furthermore, in cases of impaired tissue microcirculation, ADM has a protective effect and prevents the hypoxia of tissues [5]. 

Increased levels of serum ADM are suggestive of organ failure condition. Although ADM is considered an early diagnostics and prognostics biomarker of several diseases, such as cardiovascular disease, respiratory disease, and sepsis [6,7], the reliability and accuracy of ADM in blood are restricted due to fast clearance, plasma protein binding, and rapid degradation by proteases [8,9]. 

A solution to this problem is the dosage of a peptide product released during the ADM precursor protein cleavage and maturation, i.e., the mid-regional proadrenomedullin (MR-proADM), a fragment of 48 amino acids that splits from a proADM molecule in a 1:1 ratio with adrenomedullin. Since it is supposed that MR-proADM is a non-functional metabolic product and, for this reason, is not degraded by proteases, MR-proADM could be considered a more stable molecule than ADM with a longer half-life [10].

Several studies have reported that MR-proADM has a higher prognostic value for sepsis than biomarkers such as procalcitonin (PCT) and C-reactive protein (CRP) [11]. MR-proADM has been also shown to be a tool in patients with lower respiratory tract infections [12]. Studies with a small samples size also suggested a possible role of MR-proADM in predicting clinical outcomes, mainly mortality, in patients affected by Coronavirus disease 2019 (COVID-19). In this regard, our research group reported that MR-proADM, together with other biomarkers, is a useful tool to stratify the mortality risk in COVID-19 patients upon admission to the emergency department (ED) [13,14,15].

The infection with severe acute respiratory syndrome coronavirus 2 (SARS-CoV-2), which causes COVID-19 [16], is a systemic inflammatory disease that mostly affects the respiratory system. SARS-CoV-2 enters host cells through type 2 angiotensin-converting enzyme, which is widely expressed in a variety of organs and tissues, including the lungs, heart, kidneys, intestines, and endothelial cells [17]. 

Clinical features of COVID-19 range from asymptomatic or pauci-symptomatic conditions to severe clinical manifestations characterized by respiratory failure, requiring mechanical ventilation and support in an intensive care unit (ICU) [17]. 

Although pneumonia is the most frequent manifestation of SARS-CoV-2 infection, in several cases, it has been reported that patients with severe COVID-19 could develop acute respiratory distress syndrome (ARDS), cytokine storm, multiple organ dysfunction syndromes, and death [17,18].

During the COVID-19 outbreak, computed tomography (CT) score was widely used for its excellent sensitivity in diagnosing SARS-CoV-2-associated pneumonia [19], resulting in it becoming particularly useful for clinical diagnoses in the ED [19,20].

In order to improve patients’ risk stratification, several prognostic models combined chest CT biomarkers of COVID- 19 pneumonia severity together with clinical predictors of COVID-19 outcome. Despite the potential utility, the possible role of CT score in the risk stratification of COVID-19 patients at admission to the ED is still unclear. 

Therefore, the aim of the present study was to assess the combination of CT findings in association with specific laboratory biomarkers and with MR-proADM in the prediction of in-hospital mortality of COVID-19 patients evaluated at the triage so as to help emergency physicians in the decision-making process concerning whether patients are ruled in or ruled. 

A further goal of the present study was to evaluate whether both CT and MR-proADM results could also play a key role in predicting the correct clinical setting for COVID-19 patients, thus contributing to the optimization of hospital resources.

## 2. Materials and Methods

### 2.1. Study Design

A retrospective, observational, single-center study was conducted on 265 consecutive patients, 180 of whom were male and 85 of whom were female, with a mean age of 64 years (Table 1) and suspected COVID-19 infection admitted to the ED of Tor Vergata University Hospital, from April to December 2020.

Hypertension (43.8%), cardiovascular diseases (17%), kidney failure (15.1%), and diabetes (14%) were the principal comorbidities observed (Table 2). SARS-CoV-2 RNA was detected by real-time reverse-transcription polymerase chain reaction (RT-PCR) assay, and by radiological imaging, diagnosis of COVID-19 was performed, in accordance with the WHO interim guidelines. 

Blood culture, sputum, urine, bronchial aspirate, and/or bronchoalveolar samples were analyzed when deemed necessary.

For all patients included in the study, follow-up was performed up to 45 days. 

This study was approved by the Local Ethics Committee (approval number 87/20 on 26 May 2020) and was performed in accordance with the Declaration of Helsinki. Patient’s informed consent was avoided because of the emergency of dealing with this new disease.

### 2.2. Analysis of CT Images

Two radiologists with several years of chest-imaging experience reviewed the CT images separately and independently of the clinical and laboratory data, resolving discrepancies with comparison and consensus.

All images displayed lung (width, 1500 Hounsfield unit [HU]; level, −700 HU) and mediastinal (width, 350 HU; level, 40 HU) settings. The following chest CT findings were recorded: ground-glass opacity (GGO), crazy-paving (CP) pattern, consolidation (CO), bronchial wall thickening, traction bronchiectasis, subpleural bands, and distribution of lesions. After measuring the number of lobes affected, the predominance in the upper (above trachea bifurcation), middle (between the trachea bifurcation and the intrapulmonary vein), or lower lobes (below the level of the intrapulmonary vein) was registered.

The distribution of the axial pattern was classified as peripheral, if prevalent in the outer third of the lung, or central, if prevalent in the two inner thirds. The distribution pattern was classified as diffuse when a clearly prevalent head-to-tail or axial distribution was discernible. Furthermore, in every patient, semi-automated image-processing software was employed to assess the well-ventilated lung volume, ground-glass volume, and consolidation.

The lung parenchyma modifications were analyzed on a software-dedicated workstation using the IntelliSpace Portal 7.0 extension (Philips, Milan, Italy). Semi-automated lung segmentation and lung parenchyma analysis were obtained using CT for chronic obstructive pulmonary disease (COPD).

The ground-glass volume was depicted in the interval from −700 HU to −300 HU, and the consolidated parenchymal volume was determined in the interval from −300 HU to 40HU. In addition, the absolute volume of the altered lung volume was derived by the total lung volume software [21]. In all cases a semi-quantitative CT severity score was calculated considering the extent of anatomic involvement as follows: 0, no involvement; 1, <5% involvement; 2, 5–25% involvement; 3, 26–50% involvement; 4, 51–75% involvement; and 5, >75% involvement, using the classification proposed by Pan et al. [22] and by Kandil et al. [23]. 

The CO-RADS classification was also considered for clinical purposes (https://radiologyassistant.nl/chest/covid-19/corads-classification, accessed on 29 June 2022).

### 2.3. Blood Sample Collection

Blood samples were collected at triage admission to the ED. In order to obtain the separation of serum and plasma, blood samples were rapidly centrifuged upon arrival to the laboratory at 4500× *g* for 5 min.

Blood analyses for MR-proADM, C-reactive protein (CRP), procalcitonin (PCT), D-dimer, and lactate dehydrogenase (LDH) were performed. 

CRP (normality cut-off < 5 mg/L) and LDH (<220 IU/L) levels were measured in serum samples by an Abbott ARCHITECT c16000 (Abbott, North Chicago, IL, USA) clinical chemistry analyzer. PCT (normality cut-off < 0.5 ng/mL) was detected in serum with a BRAHMS PCT chemiluminescent microparticle immunoassay (CMIA) by Abbott ARCHITECT i2000SR instrument.

The MR-proADM level (normality cut-off < 0.55nmol/L) was assessed using a time-resolved amplified cryptate emission assay in EDTA plasma samples (TRACE BRAHMS MR-proADM Kryptor, BRAHMS AG, Hennigsdorf, Germany). D-dimer values were obtained by an ACL TOP 700 Instrument by Instrumentation Laboratory Company (Werfen, Bedford, MA, USA).

### 2.4. Statistical Analysis

In this study, two main endpoints were established: the primary endpoint was the overall in-hospital mortality; the secondary endpoint was the need for invasive mechanical ventilation (IMV). 

Continuous variables were expressed as mean (standard deviation, SD) or median (interquartile ranges, IQR), according to data distribution, and were compared using Student’s t test or the Mann–Whitney U test, where appropriate. 

Categorical variables were expressed as counts and percentages and compared using the chi-squared or Fisher’s exact tests, as appropriate.

To evaluate the correlation between CT score and biomarkers, non-parametric Spearman’s correlation coefficient (rho) was calculated.

Associations among the candidate variables and endpoints were evaluated by both univariate and multivariate Cox regression analyses, and hazard ratios were measured. Survivor vs. non-survivors and patients who needed ventilation vs. patients without ventilation were evaluated.

The discriminatory power of the variables analyzed to predict mortality was tested by means of receiver operating characteristic (ROC) curve analysis with area under the ROC curve (AUC) determination.

For the regression analysis, variables were dichotomized according to cut-off values derived during the data analysis for this study, using the Youden index arising from the ROC curve analysis.

The CT score and the biomarkers’ scores were combined using binary logistic regression. This resulted in a new variable, which was subsequently used as a test variable to run the ROC curve.

For each biomarker, sensitivity, specificity, negative and positive predictive values (NPV, PPV), negative and positive likelihood ratio (LR−, LR+), and odds ratio with CI 95% were also reported for mortality and IMV.

All statistical analyses were performed with SPSS software ver. 22. For the multivariate analysis, we used variables that were statistically significant in the univariate analysis with a *p* value < 0.01. Tests were considered significant if they yielded two-tailed *p* values < 0.05.

## 3. Results

During the observational period from April to December 2020, 74 out of 265 patients were mechanically ventilated, while 81 out of 265 patients died.

Analyzing obesity and malignancy as comorbidities, significant differences were not observed between survivors and non-survivors, whereas when considering patients with respiratory diseases, a statistical level close to significance was observed (Table 2).

All other comorbidities analyzed, including hypertension, diabetes, cardiovascular diseases, and renal diseases, showed a significant difference between the two groups considered (Table 2).

Evaluating the secondary endpoint, hypertension, diabetes, malignancy, and renal diseases show a significant difference between IMV and non-IMV, whereas obesity, cardiovascular diseases, and respiratory diseases did not show significant differences between IMV and non-IMV (Table 2).

All the biomarkers analyzed and the CT score, evaluated at ED admission, showed increased values in non-survivors when compared to survivors, as well as in IMV vs. non-IMV, always reaching a statistically significant level (Table 3).

Table 4 shows the results of the univariate Cox regression analysis. This investigation was performed in order to study the predictive role of demographic and clinical features in suspected COVID-19 patients.

Obesity and malignancy do not seem to predict 45-day mortality, whereas respiratory diseases showed a statistical level close to the significance. All the other clinical features showed significant odds ratio values.

To evaluate the possible prediction of IMV need in these patients within 28 days, all the clinical conditions reached statistical significance, except for cardiovascular disease, while respiratory disease showed a statistical level close to significance (Table 4).

All the biomarkers analyzed, as well as CT score, predicted mortality. Furthermore, as analyzed by the univariate Cox regression analysis (Table 4), all the analyzed biomarkers significantly predicted the need for IMV in patients admitted to the ED with COVID-19 infection. Furthermore, all biomarkers, except for PCT and LDH, seem to play a key role in mortality risk stratification at admission to the ED. Considering the D-dimer, it has a statistical value close to significance, whereas the CT score does not seem to be able to stratify the mortality risk in these patients. Similarly, an increase in MR-proADM level at admission was independently associated with a higher risk of the need for IMV, as well as for LDH and CT score (Table 5). 

In fact, in COVID-19 patients, CT score assessed at ED admission showed a significant, but not satisfactory, discriminating performance both for in-hospital mortality and for IMV prediction. However, when CT score was considered together with MR-proADM, the discrimination power was substantially greater for both mortality and for IMV compared to CT score alone; however, more importantly, it even greater when compared to MR-proADM alone for both outcomes, i.e., mortality and IMV need. (Figure 1 and Table 6). Moreover, considering together MR-proADM with CT score as a unique marker substantially increased specificity compared to the two single markers.

## 4. Discussion

The rapid global spread of SARS-CoV-2, the etiological agent of COVID-19, has posed enormous pressure to healthcare systems worldwide and revealed the unpreparedness of hospital emergency departments to face this unexpected pandemic, resulting in a high mortality rate. In addition, the lack of specific clinical features made it difficult to distinguish COVID-19 pneumonia from other forms of severe pneumonia [24]. Consequently, ED physicians had to evaluate a large number of patients with suspected SARS-CoV-2 infection, further stressing the already limited resources. Thus, the assessment of early risk stratification of COVID-19 patients at triage became mandatory. In this context, the utilization of prognostic tools and/or biochemical markers has been fundamental for stratifying risk and directing patients towards the right clinical pathway within the hospital.

Among the prognostic tools, computed tomography has been widely utilized in the ED to manage patients affected by SARS-CoV-2 infection, owing to its high sensitivity [25]. Accordingly, a recent study reported that CT might help emergency physicians in the clinical management of patients with negative molecular swab tests, but abnormal laboratory tests and radiological findings are highly suggestive of COVID-19 pneumonia [21]. Pan et al. proposed a CT score that correlated with disease severity and laboratory values, thus suggesting a possible predictive role of CT in patients affected by COVID-19 [22]. However, the possible role of CT score in the risk stratification of COVID-19 patients upon admission to the ED is unclear.

On the other hand, laboratory biomarkers were also useful tools for predicting the gravity of the SARS-CoV2 disease. CRP has been widely used, although its low sensitivity for community-acquired pneumonia (CAP) limited its specificity. If high CRP values (>100 mg/L) are suggestive of severe bacterial infection, lower values can be found in both viral infections and non-infectious diseases [26]. PCT is another biomarker commonly used in the ED because it increases in the presence of a bacterial infection and can be used to monitor the duration of antibiotic therapy [27] and predict the microbial etiology of pneumonia [28]. Therefore, PCT is considered a good diagnostic biomarker rather than a prognostic biomarker in patients with CAP [29]. 

A study by Christ-Crain et al. [30] demonstrated that the level of MR-proADM in plasma increased according to the severity of CAP unlike to the number of leukocytes and CRP [30]. In our study, we confirmed that MR-proADM was useful in detecting the dysfunction of the endothelium, contrasting the severity and long-term adverse outcomes of CAP.

Moreover, MR-proADM can predict poor outcomes in patients with influenza-virus-induced pneumonia [31] and can be used to stratify clinical risk in patients with CAP in the ED [32]. Recently, a predictive role of MR-proADM in COVID-19 patients with pneumonia has been reported [33,34]. In addition, our research group reported that MR-proADM, together with other biomarkers, is a useful tool for stratifying the mortality risk in COVID-19 patients at ED admission [13,14,15].

The utilization of biomarkers for COVID-19 patients has the potential to substantially optimize resources and to speed up the decision-making process of emergency physicians leading to the hospitalization of seriously ill patients only towards the more adequate care level identification.

This is particularly important since most patients infected by SARS-CoV-2 exhibit mild symptoms, whereas about 5% of them require admission to the ICU due to severe lung damage or multiorgan dysfunction [35]. Therefore, early risk stratification at the triage might reduce ED overcrowding, thus optimizing hospital resources.

Biomarkers have already been used to quickly stratify risks for patients with pneumonia and other diseases [36,37,38]. In previous studies [32,39], it has been shown that to determine the appropriate level of care at admission to the ED, as well as in CAP patients, MR-proADM can be an effective risk-stratification biomarker.

Moreover, in other studies involving hospitalized COVID-19-related pneumonia patients, it has been demonstrated that MR-proADM can play an important role in predicting patient outcomes [33,34].

As recently confirmed by our group [13], MR-proADM could be considered an effective biomarker useful in predicting death in critical patients at the ICU. Since at the ICU unfavorable outcome can happen within a few days, MR-proADM could also represent a good tool for patients’ transfer from the ED to the ICU [13]. In addition, assessed at the triage, MR-proADM can stratify the risk in terms of need of ventilation and mortality [14,15].

Since CT imaging has also been utilized in the management of patients affected by SARS-CoV-2 due to its high sensitivity [25], in this study, the risk-stratification ability of CT score alone or in association with biomarkers in patients affected by COVID-19 was explored at admission to the ED.

In line with published data [13,14,15], the median admission levels of all biomarkers evaluated showed a significantly higher value for all the endpoints considered (non-survivors vs. survivors, IMV vs. non-IMV), suggesting their predictive role in the early risk stratification of COVID-19 patients. Also, CT score confirmed higher values in non-survivors vs. survivors and in IMV vs. non-IMV, corroborating findings already reported in previous studies [21,22].

Interestingly, CT score showed a significant correlation with all the biomarkers considered, suggesting that it could also play a role in risk stratification, similarly to the other biomarkers.

However, at variance with the biomarkers evaluated, CT score did not show a significant predictive value for mortality when considering possible confounding effects of demographic and clinical features during multivariate analysis; although, during univariate analysis, it showed a significant response. On the other hand, CT score was predicted significantly for ventilation need, in both univariate and multivariate analyses. 

ROC curves analysis showed a significant, although low, AUC compared to the other biomarkers, further confirming the weak power of CT score in the risk stratification of COVID-19 patients at the ED.

Interestingly, when we consider together CT score and MR-proADM, which have been reported to be powerful in predicting mortality and the need for ventilation in patients admitted to the ED affected by COVID-19 [13,14,15], we obtain a greater discrimination power in predicting both mortality and ventilation need compared to CT score alone and to MR-proADM alone (Figure 1).

The obtained results confirm the prominent role of biomarkers as tools for early risk stratification of patients presenting to ED emergency physicians, even in the COVID-19 era.

The real novelty of this study is based on the evaluation of CT score as a predictor of mortality and ventilation need in patients infected by SARS-CoV-2 at entry to the ED. The results also show that CT score is a valuable tool in risk stratification, especially when considered together with biomarkers helpful in predicting negative outcomes.

Moreover, the analysis of the prognostic accuracy suggests that the combined utilization of different biomarkers (i.e., MR-proADM and CT score) enhances the risk-stratification power as demonstrated by the greater value of the specificity and by the confident value of NPV that is crucial for the decision-making process of emergency physicians in the management of patients affected by SARS-CoV-2 early at ED admission.

In addition, as speculated in our previous studies, the diagnostic and predictive prognostic value increased after the combined use of clinical scores and biomarkers or by utilizing a panel of biomarkers [40,41,42]. 

As with other investigations, this study has a number of limitations. The first limitation is the small number of analyzed cases since this study is a single-center, retrospective, and observational study. The other limitation is that clinical data were limited and not all laboratory parameters and characteristics were available for all patients, resulting in a few missing data. Lastly, smoking and body mass index, which represent two important confounders of MR-proADM levels, were not addressed in the adjusted regression models. Despite these limitations, our findings suggest that MR-proADM can potentially assist in identifying the most severe cases and clinical decision making in COVID-19 patients.

## 5. Conclusions

Respiratory diseases are the third cause of death in Italy, after cardiovascular diseases and neoplastic diseases. The COVID-19 pandemic has further aggravated this scenario as SARS-CoV-2 mainly affects the upper respiratory tract, causing severe respiratory disorders or exacerbating already compromised conditions. 

In the post-pandemic phase, it would be interesting to evaluate whether what we learned during the COVID-19 emergency could be used in a broader context of respiratory tract infection management, helping the physician in the decision-making process and thus contributing to the optimization of hospital resources. 

MR-proADM combined with other biomarkers and CT score seems to be a very useful tool for rapidly predicting the prognosis of SARS-CoV-2 disease in patients admitted to emergency departments.

## Figures and Tables

**Figure 1 diagnostics-13-02829-f001:**
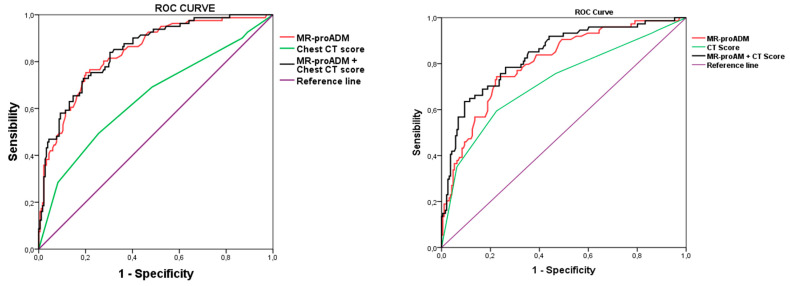
ROC curves with the association of candidate biomarkers and CT score with mortality (**left panel**) and mechanical ventilation (**right panel**). MR-proADM, mid-regional proadrenomedullin; IMV, invasive mechanical ventilation.

**Table 1 diagnostics-13-02829-t001:** Demographic and clinical parameters.

	Overall	Survivors	Non-Survivors	*p* Value	NO IMV	IMV	*p* Value
N (%)	265	184 (69.4)	81 (30.5)		191 (72)	74 (27.9)	
**Age**							
Years, mean (SD)	64 (14.4)	60.8 (14.4)	71.7 (11.1)	<0.001	62 (15.5)	68 (9.8)	<0.001
**Sex**							
Male, N (%)	180 (67.9)	121 (67.2)	59 (32.8)		121 (67.2)	59 (32.8)	0.01
				0.255			
Female, N (%)	85 (32.1)	63(74.1)	22(25.9)		70(82.4)	15 (17.6)	

**Table 2 diagnostics-13-02829-t002:** Clinical characteristic of the analyzed population.

	Overall	Survivors	Non-Survivors	*p* Value	NO IMV	IMV	*p* Value
N (%)	265	184 (69.4)	81 (30.5)		191 (72)	74 (27.9)	
**COMORBIDITIES**							
Hypertension, N (%)	116 (43.8)	65 (56.0)	51 (44.0)	<0.001	74 (63.7)	42 (36.2)	0.008
Diabetes, N (%)	37 (14.0)	16 (43.2)	21 (56.8)	<0.001	18 (48.6)	19 (51.4)	0.001
Respiratory disease, N (%)	23 (8.7)	12 (52.2)	11 (47.8)	0.060	13 (56.6)	10 (43.4)	0.080
Malignancy, N (%)	13 (4.9)	7 (53.8)	6 (46.2)	0.210	6 (46.2)	7 (53.8)	0.033
Cardiovascular disease, N (%)	45 (17.0)	22 (48.9)	23 (51.1)	0.001	30 (66.7)	15 (33.3)	0.375
Renal disease, N (%)	40 (15.1)	13 (32.5)	27 (67.5)	<0.001	15 (37.5)	25 (62.5)	0.001
Obesity, N (%)	12 (4.5)	7 (58.3)	5 (41.7)	0.390	6 (50.0)	6 (50.0)	0.081

Results expressed in percentages (%) are indicative of the proportion of patients within each group for each variable. Data are presented as mean ± standard deviation (SD), where specified. Chi-squared (χ^2^) analysis was used to determine significance among groups for categorical variables, Student’s t test was used for the variable of age. IMV, invasive mechanical ventilation.

**Table 3 diagnostics-13-02829-t003:** Biomarkers values at triage admission.

	Overall	Survivors	Non-Survivors	*p* Value	NO IMV	IMV	*p* Value
N (%)	265	184 (69.4)	81 (30.5)		191 (72)	74 (27.9)	
MR-proADM nmol/L				<0.001			<0.001
Median	0.92	0.80	1.38		0.81	1.35	
(Q1–Q3)	(0.68–1.33)	(0.60–1.01)	(1.09–2.03)		(0.61–1.06)	(0.99–1.95)	
CRP mg/L				<0.001			<0.001
Median	65.9	51.7	131		53	132	
(Q1–Q3)	(28.50–130)	(18.0–92.8)	(75.8–204.6)		(21–98)	(71–212)	
PCT ng/mL				<0.001			<0.001
Median	0.08	0.06	0.18		0.06	0.19	
(Q1–Q3)	(0.04–0.20)	(0.03–0.13)	(0.095–0.40)		(0.03–0.13)	(0.1–0.55)	
D-dimer ng/mL				<0.001			<0.001
Median	741	643	1283		666	1179	
(Q1–Q3)	(438–1446)	(408–1064)	(687–2149)		(413–1192)	(646–2047)	
LDH U/L				<0.001			<0.001
Median	358	336	456		334	486	
(Q1–Q3)	(273–489)	(265–432)	(306–589)		(257–432)	(331–593)	
CT Score	3	2	3	<0.001	2	4	<0.001
	(2–4)	(2–4)	(2–5)		(2–3)	(2.75–5)	

Data are represented as median [first quartile (Q1)–third quartile (Q3)]. The Mann–Whitney U test was used to determine significance among biomarker concentrations. MR-proADM, mid-regional proadrenomedullin; CRP, C-reactive protein; PCT, procalcitonin; LDH, lactate dehydrogenase; IMV, invasive mechanical ventilation. CT score: see text.

**Table 4 diagnostics-13-02829-t004:** Univariate Cox regression analysis for biomarkers and clinical characteristics for the primary (survivors) and secondary (IMV) outcomes. Univariate Cox regression analysis for the prediction of 45-day mortality and 28-day IMV.

	Overall (N)	Non-Survivors (N)	Cut-Off	*p* Value	HR (95% CI) (N)	IMV (N)	Cut-Off	*p* Value	HR (95% CI) (N)
Age	265	80		<0.001	1.05 (1.03–1.07)	74		0.004	1.03 (1.01–1.04)
Gender	265	80		0.27	1.32 (0.81–2.15)	74		0.018	1.98 (1.12–3.49)
Hypertension	265	80		<0.001	2.46 (1.57–3.88)	74		0.013	1.80 (1.13–2.85)
Diabetes	265	80		<0.001	2.63 (1.58–4.36)	74		0.001	2.46 (1.46–4.15)
Respiratory disease	265	80		0.078	1.78 (0.94–3.37)	74		0.069	1.85 (0.95–3.62)
Malignancy	265	80		0.120	1.93 (0.84–4.45)	74		0.023	2.47 (1.13–5.38)
Cardiovascular disease	265	80		<0.001	2.41 (1.49–3.92)	74		0.40	1.28 (0.73–2.25)
Renal disease	265	80		<0.001	4.25 (2.65–6.81)	74		<0.001	3.87 (2.38–6.30)
Obesity	265	80		0.50	1.36 (0.55–3.36)	74		0.044	2.36 (1.02–5.44)
MR-proADM (nmol/L)	265	80	1.105	<0.001	7.3 (4.39–12.2)	74	1.105	<0.001	6.03 (3.57–10.18)
CRP (mg/L)	265	80	95.5	<0.001	5.72 (3.52–9.3)	74	63	<0.001	5.55 (2.99–10.3)
PCT (ng/mL)	265	80	0.095	<0.001	4.67 (2.81–7.76)	74	0.095	<0.001	5.41 (3.11–9.42)
D-dimer (ng/mL)	265	80	985.5	<0.001	3.99 (2.51–6.35)	74	981.5	<0.001	3.23 (2.01–5.2)
LDH (U/L)	265	80	439.5	<0.001	3.38 (2.12–5.13)	74	437.5	<0.001	3.93 (2.47–6.26)
CT Score	265	80	>3	<0.001	2.37 (1.53–3.7)	74	>3	<0.001	3.70 (2.33–5.90)

HR, hazard ratio; CI, confidence interval; CRP, C-reactive protein; MR-proADM, mid-regional proadrenomedullin; PCT, procalcitonin; LDH, lactate dehydrogenase; CT score, Computed Tomography score; IMV, Invasive Mechanical Ventilation.

**Table 5 diagnostics-13-02829-t005:** Multivariate Cox regression analysis pooling together biomarkers and clinical characteristics for the primary (survivors) and for the secondary (IMV) outcomes. Multivariate Cox regression analysis for the prediction of 45-day mortality and 28-day IMV.

	Overall (N)	Non-Survivors (N)	*p* Value	HR (95% CI)	IMV (N)	*p* Value	HR (95% CI)
Age	265	80	0.06	1.02 (0.99–1.05)	74	0.47	0.99 (0.97–1.02)
Gender	265	80			74	0.77	1.10 (0.59–2.07)
Hypertension	265	80	0.85	1.05 (0.64–1.73)	74	0.81	1.07 (0.63–1.80)
Diabetes	265	80	0.63	1.16 (0.64–2.08)	74	0.12	1.6 (0.89–2.87)
Respiratory disease	265	80	0.18	1.60 (0.81–3.14)	74	0.07	1.95 (0.94–4.0)
Malignancy	265	80			74	<0.001	5.59 (2.44–14.42)
Cardiovascular disease	265	80	0.10	1.63 (0.91–2.93)	74		
Renal disease	265	80	0.07	1.61 (0.97–2.70)	74	0.03	1.81 (1.06–3.10)
Obesity	265	80			74	0.44	1.45 (0.56–3.72)
MR-proADM (nmol/L)	265	80	0.001	2.84 (1.57–5.14)	74	0.003	2.72 (1.41–5.24)
CRP (mg/L)	265	80	<0.001	3.10 (1.79–5.39)	74	0.098	1.62 (0.91–2.88)
PCT (ng/mL)	265	80	0.50	1.25 (0.66–2.35)	74	0.09	1.82 (0.91–3.62)
D-dimer (ng/mL)	265	80	0.059	1.71 (0.98–3.0)	74	0.47	1.23 (0.71–2.12.45)
LDH (U/L)	265	80	0.161	1.49 (0.85–2.59)	74	0.015	2.06 (1.15–3.67)
CT score	265	80	0.92	0.97 (0.57–1.66)	74	0.016	2.13 (1.14–3.49)

HR, hazard ratio; CI, confidence interval. Age, hypertension, diabetes, respiratory disease, malignancy, cardiovascular disease, and renal disease were used as adjusting variables within the multivariate Cox regression analysis for the prediction of 45-day mortality. Age, gender, hypertension, diabetes, respiratory disease, and renal disease were used as adjusting variables within the multivariate Cox regression analysis for the prediction of 28-day IMV. Age, hypertension, and renal disease were used as adjusting variables within the multivariate Cox regression analysis for the prediction of 28-day IMV. MR-proADM, mid-regional proadrenomedullin; CRP, C-reactive protein; PCT procalcitonin; LDH, lactate dehydrogenase; CT score, computed tomography score.

**Table 6 diagnostics-13-02829-t006:** Prognostic accuracy of biomarkers plus CT score.

	Outcome	AUC (95% CI)	Cut-Off	Sensitivity (95% CI)	Specificity (95% CI)	PPV (95% CI)	NPV (95% CI)	LR+ (95% CI)	LR− (95% CI)	OR (95% CI)
MR-proADM	Mortality	0.839 (0.79–0.89)	1.105	0.75 (0.64–0.84)	0.79 (0.72–0.85)	0.61 (0.54–0.68)	0.88 (0.83–0.92)	3.55 (2.62–4.82)	0.31 (0.21–0.46)	11.34 (6.12–21.01)
	IMV	0.803 (0.74–0.86)	1.105	0.74 (0.63–0.84)	0.76 (0.70–0.82)	0.55 (0.48–0.62)	0.89 (0.84–0.92)	3.15 (2.36–4.21)	0.34 (0.23–0.50)	9.39 (5.05–17.45)
CT score	Mortality	0.646 (0.57–0.72)	>3	0.49 (0.38–0.61)	0.74 (0.67–0.80)	0.46 (0.38–0.54)	0.77 (0.73–0.81)	1.89 (1.36–2.63)	0.68 (0.54–0.86)	2.76 (1.6–4.8)
	IMV	0.719 (0.65–0.79)	>3	0.60 (0.47–0.71)	0.77 (0.70–0.83)	0.50 (0.42–0.58)	0.83 (0.79–0.87)	2.58 (1.87–3.56)	0.53 (0.40–0.70)	4.9 (2.76–8.69)
MR-proADM + CT score	Mortality	0.847 (0.80–0.90)	1.105 >3	0.73 (0.62–0.82)	0.81 (0.75–0.86)	0.63 (0.55–0.70)	0.87 (0.83–0.91)	3.83 (2.8–5.3)	0.34 (0.2–0.5)	11.42 (6.19–21.07)
	IMV	0.837 (0.78–0.89)	1.105 >3	0.64 (0.52–0.74)	0.91 (0.86–0.94)	0.72 (0.62–0.81)	0.87 (0.83–0.90)	6.74 (4.2–10.8)	0.40 (0.30–0.55)	16.73 (8.49–32.96)

AUC analysis for 45-day mortality prediction and for 28-day IMV prediction of study population. *p* value: differences between areas of each biomarker vs. MR-pro-ADM. Cut-off derived from ROC (receiver operating characteristic) using the Youden index. MR-proADM (nmol/L), mid-regional proadrenomedullin; CT score, computed tomography score; IMV, invasive mechanical ventilation.

## Data Availability

The data of this study are available in this article.

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
