# Peer review of "Correlation between Chest Computed Tomography Score and Laboratory Biomarkers in the Risk Stratification of COVID-19 Patients Admitted to the Emergency Department"

_diagnostics, 2023, doi:10.3390/diagnostics13172829_

Round 1
Reviewer 1 Report
The manuscript of D'Agostini C et al. entitled "Correlation between Chest CT Score and Laboratory Biomarkers in the Risk Stratification of COVID-19 Patients Admitted to the Emergency Department" corresponds to a retrospective, observational work in a single-center study. This work has Scientific Soundness, however some analyzes are missing.
· = Authors should indicate all acronyms the first time they appear, such as Community-acquired pneumonia (CAP)
· =The objective in the abstract is not consistent with the introduction and conclusion.
1. " Therefore, the aims of this study were to assess the CT findings alone or in association with MR-proADM in the prediction of in-hospital mortality of COVID-19 patients evaluated at the triage and to evaluate whether CT and MR-proADM results can play a key role in predicting also the correct clinical setting for these patients." VS
2. "MR-proADM appears to be a useful tool for predicting the prognosis of certain pathophysiological conditions in patients presenting to the emergency department" VS
3. “A further goal of the present study was to evaluate whether both CT and MR-proADM results can play a key role in predicting also the correct clinical setting for these patients thus contributing to optimize the hospital resources”
· = In Analysis of CT images. The authors must indicate the CO-RADS score (https://radiologyassistant.nl/chest/covid-19/corads-classification)
· = Due to the results of the CT score in Table 6, it is suggested to analyze with other CT scores.
1. Kandil, Omneya, et al. "Prognostic and discriminatory abilities of imaging scoring systems in predicting COVID‐19 adverse outcomes." iRADIOLOGY 1.2 (2023): 128-140.
2. Lieveld AWE, et al. Chest CT in COVID-19 at the ED: Validation of the COVID-19 Reporting and Data System (CO-RADS) and CT Severity Score: A Prospective, Multicenter, Observational Study. Chest. 2021;159(3):1126-1135. doi: 10.1016/j.chest.2020.11.026.
3. CO-RADS score https://radiologyassistant.nl/chest/covid-19/corads-classification
· = In Results.-How many patients died in the group that received mechanical ventilation?
· = In Table 3, what type of CT score do, the authors report (was by Pan et al.)?
· =In figure 1. Show the ROC curves comparing MR-proadrenomedullin + CRP + Chest CT score. Also MR-proadrenomedullin + D-dimer + Chest CT score.
· =Please indicate limitations of the work.
Author Response
The manuscript of D'Agostini C et al. entitled "Correlation between Chest CT Score and Laboratory Biomarkers in the Risk Stratification of COVID-19 Patients Admitted to the Emergency Department" corresponds to a retrospective, observational work in a single-center study. This work has Scientific Soundness, however some analyzes are missing.
- = Authors should indicate all acronyms the first time they appear, such as Community-acquired pneumonia (CAP)
Thank you. All acronyms were indicated in the manuscript.
- =The objective in the abstract is not consistent with the introduction and conclusion.
Thank you for the comment. We modified the abstract consistently and reviewed the text to improve and uniform the content of the work. Please, see Introduction pag.2 line 48 and Conclusion pag.12 line 10 and other consideration point-to-point review.
- " Therefore, the aims of this study were to assess the CT findings alone or in association with MR-proADM in the prediction of in-hospital mortality of COVID-19 patients evaluated at the triage and to evaluate whether CT and MR-proADM results can play a key role in predicting also the correct clinical setting for these patients." VS
The real goal of the present study is not a challenge among different biomarkers in the risk stratification in COVID-19 patients in the Emergency Department, but rather we intend to evaluate how different biomarkers combined together could improve the risk stratification analysis. Finally, the take home message, as supported by the results of our study, should be that using different biomarkers together improve the power of predicting the risk in patients affected by COVID-19 when evaluated early at the Emergency Department admission.
- "MR-proADM appears to be a useful tool for predicting the prognosis of certain pathophysiological conditions in patients presenting to the emergency department" VS
It has been shown in previous papers (reported in the references) and confirmed in the present study with the novelty that we can obtain better results by combining the diagnostic information from different biomarkers.
- “A further goal of the present study was to evaluate whether both CT and MR-proADM results can play a key role in predicting also the correct clinical setting for these patients thus contributing to optimize the hospital resources”
Results of our study show that MR-proADM is able to stratify
In the discussion section (page 12, 11st paragraph) we state “Interestingly, when we consider together CT score and MR-proADM…… we obtain a greater discrimination power in predicting both mortality and ventilation need as compared to CT score alone but also to MR-proADM alone.” The need of ventilation can help the emergency physician to select the adequate hospital setting for the patient.
- = In Analysis of CT images. The authors must indicate the CO-RADS score (https://radiologyassistant.nl/chest/covid-19/corads-classification)
Done. Thank you.
- = Due to the results of the CT score in Table 6, it is suggested to analyze with other CT scores.
We used Pan et al. CT score (the same of Kandil et al) to correlate CT and laboratory tests with a possible predictive role in patients affected by COVID-19 disease. This statement is enclosed in the Discussion of the draft (page 10, paragraph 2). Another reason for this choice is the possible quantification of the lung damage in percentage to better analyze and compare data. For this aim, we used a semi-automated image processing described in the text (page 4; lines 11-20). Finally, CT score is simple to apply and is not time consuming, thus more adequate for the setting of the Emergency Department. We added also the Kandil’s reference, #[23], in the manuscript.
Kandil, Omneya, et al. "Prognostic and discriminatory abilities of imaging scoring systems in predicting COVID‐19 adverse outcomes." iRADIOLOGY 1.2 (2023): 128-140.
- Lieveld AWE, et al. Chest CT in COVID-19 at the ED: Validation of the COVID-19 Reporting and Data System (CO-RADS) and CT Severity Score: A Prospective, Multicenter, Observational Study. Chest. 2021;159(3):1126-1135. doi: 10.1016/j.chest.2020.11.026.
- CO-RADS score https://radiologyassistant.nl/chest/covid-19/corads-classification
- = In Results.-How many patients died in the group that received mechanical ventilation?
The number of deaths in the Invasive Mechanical Ventilation (IMV) group is out the goals of the study. Two main endpoints were established: the primary was the overall in-hospital mortality; the secondary endpoint was the need of IMV. Therefore, we did not calculate how many patients died in the IMV group.
- = In Table 3, what type of CT score do, the authors report (was by Pan et al.)?
Done. Thank you.
- =In figure 1. Show the ROC curves comparing MR-proadrenomedullin + CRP + Chest CT score. Also MR-proadrenomedullin + D-dimer + Chest CT score.
The final aim of the present study was to assess the combination of CT results in association with specific laboratory biomarkers and, particularly, with MR-proADM, in the prediction of in-hospital mortality of COVID-19 patients evaluated at the triage in order to help the emergency physician in the decision-making concerning the rule-in or rule-out of these patients. The role of D-dimer as a COVID-19 biomarker has been widely demonstrated and it is behind the goals of the present study. For this reason, we did not calculate the ROC curves for MR-proadrenomedullin + D-dimer + Chest CT score.
- =Please indicate limitations of the work.
We have added a paragraph at the end of the text.

Reviewer 2 Report
What is interesting is that the CT score seems to have added marginal diagnostic value to the MR-proADM biomarker. Please address the following comments:
1. Figure 1 caption mentions "upper" and "lower" panels, but the panels are side by side.
2. Mathematically, how are the CT score and the biomarker scores combined to create the last part of Table 7?
3. In Figure 1, the claim is that the black curves have more area under the curve than the red curves. What is the statistical significance of the difference? The difference in the right panel looks convincing, but the difference in the left panel seems minimal. The same seems to be the case with the mortality data in Table 7. Some analysis of the statistical significance will be helpful.
Needs some grammatical corrections.
Author Response
What is interesting is that the CT score seems to have added marginal diagnostic value to the MR-proADM biomarker. Please address the following comments:
- Figure 1 caption mentions "upper" and "lower" panels, but the panels are side by side.
Thank you. The caption was updated correctly.
- Mathematically, how are the CT score and the biomarker scores combined to create the last part of Table 7?
Thanks to the reviewer for this comment which allows us to better clarify this point in the manuscript. In the statistical analysis subheading, we have added the following sentence to clarify the issue: “The CT score and the biomarkers’ scores have been combined using a binary logistic regression. This resulted in a new variable which was subsequently used as test variable to run the ROC curve.”
- In Figure 1, the claim is that the black curves have more area under the curve than the red curves. What is the statistical significance of the difference? The difference in the right panel looks convincing, but the difference in the left panel seems minimal. The same seems to be the case with the mortality data in Table 7. Some analysis of the statistical significance will be helpful.
Thank you for this comment. Actually, we did not perform the statistical analysis of the difference between the MR-proADM values alone vs MR-proADM+CT score, because it was not the goal of this study to compare these biomarkers. The message of the present study is that the CT score alone offers less information as compared to the MR-proADM level plus the CT score.

Reviewer 3 Report
Major
COVID-19 patients have a variety of complications including venous thromboembolism, pulmonary embolism, disseminated intravascular coagulations. The involvement of coagulation and inflammatory markers on the prognosis have been extensively examined.
Mid-Regional pro-Adrenomedullin (MR-proADM) is an inflammatory biomarker that improves the prognostic assessment of patients with sepsis, septic shock and organ failure (.Respir Res 23, 221 (2022). ). It has been reported that MR-proADM in combination with age and CRP or with the patient’s SOFA score could identify patients at low risk where outpatient treatment may be safe.
Further. MR-proADM) could be considered a useful tool for stratifying the mortality risk in COVID-19 patients upon entry into the Emergency Department (ED).
In addition to the biochemical markers, the imaging methods including CT is an important modality for diagnosis diagnostic accuracy may be improved by combining clinical evidence with results from chest computed tomography (CT) and RT-PCR. However, the possible role of CT in the risk stratification of COVID-19 patients at the admission in the ED is less clear.
In the current study, the authors have assessed the CT findings alone or in association with MR-proADM in the prediction of in-hospital mortality of COVID-19 patients.
The results indicated that CT score is particularly effective when utilized together with the MR-proADM level in the risk stratification of COVID-19 patients admitted to the ED.
All the biomarkers and the CT score, evaluated at the ED admission, showed an increased values in non-survivors when compared to survivors as well as inf invasive mechanical ventilation (IMV) vs non-IMV.
In their conclusion, MR-proADM in combination with CT examination seems to be a useful tool for predicting the prognosis of certain pathophysiological conditions in patients presenting to emergency departments.
The authors should discuss the significant utility of MR-proADM as a severity of COVID-19 compared with that of other biochemical markers.
Author Response
COVID-19 patients have a variety of complications including venous thromboembolism, pulmonary embolism, disseminated intravascular coagulations. The involvement of coagulation and inflammatory markers on the prognosis have been extensively examined.
Mid-Regional pro-Adrenomedullin (MR-proADM) is an inflammatory biomarker that improves the prognostic assessment of patients with sepsis, septic shock and organ failure (.Respir Res 23, 221 (2022). ). It has been reported that MR-proADM in combination with age and CRP or with the patient’s SOFA score could identify patients at low risk where outpatient treatment may be safe.
Further. MR-proADM) could be considered a useful tool for stratifying the mortality risk in COVID-19 patients upon entry into the Emergency Department (ED).
In addition to the biochemical markers, the imaging methods including CT is an important modality for diagnosis diagnostic accuracy may be improved by combining clinical evidence with results from chest computed tomography (CT) and RT-PCR. However, the possible role of CT in the risk stratification of COVID-19 patients at the admission in the ED is less clear.
In the current study, the authors have assessed the CT findings alone or in association with MR-proADM in the prediction of in-hospital mortality of COVID-19 patients.
The results indicated that CT score is particularly effective when utilized together with the MR-proADM level in the risk stratification of COVID-19 patients admitted to the ED.
All the biomarkers and the CT score, evaluated at the ED admission, showed an increased values in non-survivors when compared to survivors as well as inf invasive mechanical ventilation (IMV) vs non-IMV.
In their conclusion, MR-proADM in combination with CT examination seems to be a useful tool for predicting the prognosis of certain pathophysiological conditions in patients presenting to emergency departments.
The authors should discuss the significant utility of MR-proADM as a severity of COVID-19 compared with that of other biochemical markers.
In the discussion section (page 12, 4th paragraph), we state “…the analysis of the prognostic accuracy suggests that the combined utilization of different biomarkers (i.e., MR-proADM and CT score) enhances the risk stratification power as demonstrated by the greater value of the specificity and by the confident value of NPV that is crucial for the decision-making process of the emergency physician in the management of patients affected by SARS-CoV-2 early at the ED admission.”
The real goal of the present study is not a challenge among different biomarkers in the risk stratification in COVID-19 patients admitted to the Emergency Department, but rather we intend to evaluate how different biomarkers combined together could improve the risk stratification analysis. Finally, the message to take home should be that using different biomarkers together improve the power of predicting the risk in patients affected by COVID-19 when evaluated early at the Emergency Department admission.
Thank you.

Round 2
Reviewer 1 Report
1. Please include the meaning of any acronyms the first time they appear, e.g. AUROC
2. Limitations are not conclusions, please change this in the last section of the discussion.